# Somatic Marker Production Deficits Do Not Explain the Relationship between Psychopathic Traits and Utilitarian Moral Decision Making

**DOI:** 10.3390/brainsci10050303

**Published:** 2020-05-15

**Authors:** Shawn E. Fagan, Liat Kofler, Sarah Riccio, Yu Gao

**Affiliations:** 1Indiana University Bloomington, Bloomington, IN 47405, USA; 2The Graduate Center, City University of New York, New York, NY 10016, USA; lkofler@gradcenter.cuny.edu (L.K.); yugao@brooklyn.cuny.edu (Y.G.); 3Illinois School of Professional Psychology (ISPP) at National Louis University, Chicago, IL 60603, USA; stariccio@landstromcenter.com; 4Brooklyn College, City University of New York, Brooklyn, NY 11210, USA

**Keywords:** moral decision-making, utilitarianism, somatic marker hypothesis, psychopathy, harm aversion

## Abstract

In moral dilemma tasks, high levels of psychopathic traits often predict increased utilitarian responding—specifically, endorsing sacrificing one person to save many. Research suggests that increased arousal (i.e., somatic marker production) underlies lower rates of utilitarian responding during moral dilemmas. Though deficient somatic marker production is characteristic of psychopathy, how this deficit affects the psychopathy–utilitarian connection remains unknown. We assessed psychopathic traits in undergraduates, as well as behavioral performance and skin conductance level reactivity (SCL-R; a measure of somatic marker production) during a moral dilemma task. High psychopathic traits and low SCL-R were associated with increased utilitarian decisions in dilemmas involving direct personal harm. Psychopathic traits were unrelated to SCL-R, nor did SCL-R mediate the relationship between psychopathy and utilitarianism. The present study did not find evidence that somatic marker production explains the connection between utilitarianism and psychopathy in a college population. Further research is necessary to identify the neural mechanisms relating psychopathy and moral decision-making in nonclinical samples.

## 1. Introduction

From the historical case study of Phineas Gage to the HBO documentary on Robert Durst, there remains an ongoing fascination with psychopathic individuals for both psychologists and laypersons alike. Psychopathy is a personality disorder characterized by a lack of empathy, callousness, and antisocial behavior [1,2,3]. Though past research has traditionally supported a genetic etiology of psychopathy, there is growing evidence that social factors contribute to the development of the disorder as well [4]. Traditionally, psychopathy is differentiated into primary and secondary subtypes. While primary psychopathy is associated with low anxiety, hypoarousal, and “distinct anomalies in cognitive and attentional functioning,” secondary psychopathy is related to “hyperarousal, emotion dysregulation, and high responsivity to motivational stimuli” [5] (p. 2). Individuals are categorized as having primary versus secondary psychopathy by measuring neurotic anxiety [5].

Those with high levels of psychopathic traits often display stereotypically amoral decision making [6,7]. Many studies evaluate moral decision making using sacrificial moral dilemmas, such as variants of the classic trolley problem [8], in which one must choose between flipping a switch to divert a speeding trolley from a track with five people to a track with one person or doing nothing and letting the trolley remain on the track, allowing the five to die by default. Individuals in this impersonal hypothetical situation typically opt to flip the switch to save five, endorsing a utilitarian view of moral decision making. In the footbridge problem, an alternate version of the trolley scenario, one must push a stranger from a footbridge into the path of the trolley to save the five people. In the second dilemma, the choice involves personally inflicting harm on another individual. Thus, participants are reluctant to push the stranger onto the tracks [9], though it remains the optimal utilitarian option.

The tension between personal and impersonal moral dilemmas is explained by Josh Greene’s canonical dual-process theory of moral judgment [10]. The dual-process theory broadly proposes that utilitarian processing engages brain areas underlying cool executive functioning, while moral deliberation that involves personal harm is influenced by emotion-specific psychological and neural structures underlying empathic concern and perspective-taking [2,11]. These processes are independent, meaning that questions involving personal moral dilemmas increase emotional engagement rather than decrease utilitarian or cognitive processing [11].

The increased emotional feedback that individuals experience when confronted with a personal moral dilemma where they are asked to consider personally harming another human being is often accompanied by a sense of discomfort that is associated with an aversion to causing harm. This concept of “harm aversion” is grounded in theories involving punishment learning and fear conditioning [12,13] and can partially be explained by the Somatic Marker Hypothesis (SMH). The SMH proposes that during the learning process, autonomic or visceral responses to negative events, also known as somatic markers, become integrated in a neural network such that, over time, simply anticipating a negative event can activate effector structures such as the brainstem, amygdala, and ventromedial prefrontal cortex [14,15]. These effector structures then generate an unpleasant gut feeling—as though the person had engaged in the harmful or destructive activity in real time and were experiencing the negative consequences of their actions (see Figure 1 for a detailed diagram). Brain imaging findings support the SMH by showing concomitant activation in the aforementioned effector structures during the Iowa gambling task [16] and proportionally strong activity in the right insula (an area critically involved in negative emotion processing including disgust) following risky responding [17].

Somatic markers manifest in the autonomic nervous system [18] and can be measured via electrodermal activity (EDA; also known as the skin conductance level or SCL), a physiological index of sympathetic nervous system activity [19,20]. Overall, somatic markers are adaptive and beneficial in that they encourage neurotypical individuals to avoid risky decisions that have a high likelihood of a negative outcome [14,20]. They also have the capacity to encourage prosocial behavior; for example, changes in SCL in response to watching another person experience pain were positively correlated with subsequent helping behavior, even when such behavior was costly to the person [21]. Additionally, execution of harmless actions that participants knew to be benign in the context of an experiment (but would cause harm in real life) increased physiological activity [22].

Somatic marker production is an integral process of punishment learning [19]. This is particularly relevant to the present study given evidence that psychopaths (principally those with high levels of primary psychopathic traits) show reduced skin conductance responses (SCRs) during punishment learning [23], although some studies have failed to find a significant relationship between psychopathic traits and skin conductance levels [24,25,26]. Reduced SCRs suggest an underlying disruption to one or more pathways in the neural circuitry of the SMH, ultimately affecting the production of anticipatory physiological markers. An inability to experience an unpleasant “gut feeling” in response to negative stimuli, particularly people in distress, has detrimental consequences, including diminished harm aversion. Consistent with this argument, prior studies showed that in a moral dilemma task, participants produced larger SCRs during the contemplation period in personal versus impersonal scenarios [27] and that larger SCRs when contemplating moral dilemmas are related to fewer utilitarian decisions [28]. 

Previous studies have reported that high levels of psychopathic traits are associated with greater endorsement of utilitarian responses in both impersonal and personal moral dilemmas [2,29,30,31]. In particular, Koenigs et al. (2012) [30] showed that incarcerated psychopaths with low anxiety (characteristic of primary psychopathy) were more likely than highly anxious psychopaths to endorse utilitarian decisions in personal moral dilemmas. Kahane et al. (2015) similarly found that utilitarian decisions in personal moral dilemmas were associated with primary psychopathy in a non-clinical sample [32].

Endorsement of the utilitarian option in a sacrificial moral dilemma does not necessarily signify the adoption of a fixed deontological or utilitarian viewpoint. There are two paths that might lead to utilitarian decision making in sacrificial personal moral dilemmas [11]. One route is the traditional utilitarian approach, which focuses on the outcome of the action, i.e., the greatest number of lives saved. The second route is driven by attenuated harm aversion, i.e., one is not averse to personally harming another [12]. Historically, the first route (concern with the greater good) was thought to be the main driving force behind utilitarian decision making; however, Kahane and colleagues (2015) [28] proposed that decisions in sacrificial moral dilemmas are principally driven by empathic concern and/or harm aversion [28,33]. This suggests that reduced harm aversion may actually be the underlying mechanism of utilitarian decision making [13,14]. This is further supported by studies that found that utilitarian decision making was associated with antisocial traits [11,15] and reduced empathy [32,34], characteristics that are not congruent with concern for the greater good and maximization of lives saved. These findings suggest that for individuals with higher levels of psychopathic traits, attenuated harm aversion is a more plausible underlying factor of increased utilitarian decisions in sacrificial personal moral dilemmas. This stands in contrast to impersonal moral dilemmas, where harm aversion is likely not at the forefront of people’s minds.

Additionally, a large body of literature shows a consistent relationship between psychopathic traits and impaired reinforcement learning and overall somatic marker production deficiencies [14,23,35,36]. Specifically, Van Honk et al. (2002) [23] found that individuals with high levels of psychopathic traits failed to develop somatic markers during the Iowa gambling task, leading them to make more disadvantageous, risky choices. Additional studies employing Pavlovian fear conditioning paradigms showed that individuals with high levels of psychopathic traits did not exhibit a differential startle response or increased SCL to a conditioned noxious stimulus despite displaying cognitive awareness of the unconditioned/conditioned stimuli relationship, which provides further evidence of a deficit in affective associative learning in psychopaths [35,36]. See Figure 2 for a conceptual diagram detailing the effect that psychopathy has on contingency learning.

Despite the evidence presented above, no study has directly investigated the extent to which physiological activity explains the relationship between moral decision making and psychopathic traits. Given the three-way association between somatic marker production deficiency, psychopathy, and utilitarian responding, we conducted a study in college students that evaluated physiological reactivity and behavioral performance on a traditional moral dilemma task as well as psychopathic traits. Consistent with prior literature and theoretical models, we hypothesized that (1) higher levels of primary psychopathic traits would be related to increased utilitarian responding during both personal and impersonal moral dilemma tasks. In line with past research on psychopathy and the SMH, we also hypothesized that (2) decreased somatic responses (i.e., low SCL) when considering harming another human being in the personal moral dilemmas would correspond to increased utilitarian choices, in addition to correlating with higher levels of primary psychopathic traits. Finally, we hypothesized that (3) reduced autonomic reactivity during the moral task would mediate the relationship between primary psychopathic traits and utilitarian decision-making in personal moral dilemmas.

## 2. Materials and Methods

Participants were 100 students from Brooklyn College (mean age = 22.06, SD = 6.47, 66% female) of diverse ethnic backgrounds (31% Caucasian, 20% Black, 8% Latino, 28% Asian, 7% other, and 27% unknown). They completed a moral dilemma task, a self-report questionnaire assessing psychopathic traits, and a short demographic survey querying age, gender, and ethnic background. Participants were in the laboratory for about one hour. The protocol was approved by the university Institutional Review Board (no. 309020, approved 9 January 2013) and conducted in accordance with the Declaration of Helsinki. Students gave their informed consent prior to the study and received course-credit for their participation. 

Psychopathic traits were assessed using the Levenson Self-Report Psychopathy Scale (LSRP) [36]. The LSRP is a self-report measure consisting of 26 items, 16 of which measure primary psychopathy (e.g., In today’s world, I feel justified in doing anything I can get away with to succeed) and 10 that measure secondary psychopathy (e.g., I find myself in the same kinds of trouble, time after time). Items are scored from 1 (Disagree Strongly) to 4 (Agree Strongly). Higher scores indicate higher levels of psychopathic traits, and Cronbach’s alpha for the LSRP total score was 0.77, 0.79 for the primary psychopathy subscale, and 0.58 for the secondary psychopathy subscale in the current sample. The relatively low reliability score for the secondary psychopathy subscale is comparable to that of other studies (for a summary, see Miller, Gaughan, and Pryor, 2008) [37] and deemed acceptable for a sub-scale with so few items [38].

The computerized moral dilemma task consisted of 15 dilemmas (four non-moral/neutral, four impersonal moral, and seven personal moral). Moral dilemmas were selected from a previously published set [10,39] and were presented to participants in random order (see Appendix B for the specific dilemmas used). Each dilemma was presented to participants and remained on the screen for a total of 45 s, followed by a question prompt specific to the scenario—e.g., “Would you put false information on your resume?”—which remained on the screen for 15 s. Participants pressed either the yes or no button when they made their decision. A blank screen followed for 15 s before the presentation of the next dilemma. Percentages of “yes” responses were calculated to derive separate utilitarian response rates for personal moral and impersonal moral dilemmas [2]. The number of times a participant responded “yes” to either a personal or impersonal moral dilemma was divided by the total number of moral dilemmas to create an overall utilitarian response rate for each participant.

Skin conductance was recorded using a BIOPAC 150 system (BIOPAC Systems Inc, Goleta, CA, USA) and data were analyzed offline using AcqKnowledge 4.2 software (BIOPAC Systems Inc, Goleta, CA, USA). All electrodermal activity underwent a low pass filter with a sampling rate of 200. Under continuous physiological recording, participants completed two two-minute rest periods (at the onset and conclusion of physiological recording) and a moral dilemma task, with short breaks in between.

For psychophysiological data quantification, each moral dilemma trial was divided into three epochs: the first 15 s just prior to the presentation of the dilemma (baseline), the next 45 s while the dilemma was on the screen (the contemplation period), and the final 15 s during the question prompt (decision period). Previous studies have been inconsistent regarding which period of time in a trial they select as reflective of the SCL activity that directly affects a participant’s response [28,40,41,42]. For the purposes of this study, we averaged the skin conductance data from the 45-second contemplation period and subtracted the mean SCL of the baseline period preceding the dilemma to create a reactivity score (SCL-R).

A post-hoc power analysis using the software package G* Power 3.1 (Heinrich Hein University, Duesseldorf, Germany) [43] showed that a sample size of 100 was sufficient to detect a medium to large effect size (power > 0.80), though not to detect a small effect. Missing data due to incomplete behavioral (*n* = 2) and questionnaire responses (*n* = 4) were deleted pairwise, as was missing physiological data due to equipment malfunction or experimenter error (*n* = 6). Descriptive statistics and Pearson’s correlations were computed using SPSS 26 (IBM Corporation, Armonk, NY, USA). Correlations measured the relationship between psychopathic traits, percentage of utilitarian responses, and psychophysiological data from the moral dilemma task. We then used a mediation analysis to investigate the extent to which psychophysiological indices of arousal (SCL-R during moral contemplation) mediated the relationship between psychopathy and utilitarian response rate. 

A simple mediation model with a single mediator consists of three pathways: path *a*, the relationship between the predictor and the mediator; path *b*, the relationship between the mediator and the outcome measure; and path *c*, the relationship between the predictor and the outcome measure. We used the program *brms* (an R package for Bayesian multilevel modeling) [44,45] and the sjstats R package [46] to evaluate the effects of the mediator on the relationship between psychopathic traits and utilitarian response rates. Full mediation occurs if the relationship between psychopathic traits and utilitarian response rate (path *c’*) becomes non-significant upon adding the mediator to the model; partial mediation would occur if the relationship between psychopathic traits and utilitarian response rate remained statistically significant inclusive of the mediator in the model, though the strength of the relationship would be significantly lessened [47,48]. The *brms* package evaluates mediation similarly; estimated values represent the posterior distribution means which are analogous to regression coefficients [49]. Rather than confidence intervals, however, *brms* analysis provides 95% credible intervals, which have a 95% chance of containing the true distribution mean and represent uncertainty around the estimated distribution mean [49,50]. The indirect effect is determined by the hypothesis function, which uses an evidence ratio of Bayes Factors to test the null hypothesis (that the product of the *a* and *b* mediation pathways equals zero) against the alternative [44].

We also calculated Bayes Factors to evaluate the evidence in favor of the alternative hypothesis (that SCL-R mediates the relationship between psychopathic traits and utilitarian response rates), using R code developed by Baguley and Kaye (2010) [51]. In frequentist methods, one uses a *p*-value to conclude whether there is sufficient evidence to reject the null hypothesis in favor of the alternative hypothesis, but not if there is evidence in favor of the null [52]. Bayes Factors, alternatively, compute the odds of the null hypothesis “being true” or the alternative hypothesis “being true” [53] (p. 85). A Bayes Factor value of 1 indicates no evidence in favor of either the null or the alternative hypothesis, between 1/3–1 provides some evidence in favor of the null, between 1/10–1/3 provides moderate evidence of the null, and values smaller than 1/10 indicate strong to extremely strong evidence in favor of the null [53,54,55]. Values between 1–3 provide some evidence in favor of the alternative, 3–10 offer moderate evidence in favor of the alternative, and values over 10 indicate strong to extremely strong evidence in favor of the alternative hypothesis [53,54,55]. Data and R code are available in the Appendix A.

## 3. Results

### 3.1. Correlations

All variables were normally distributed (including SCL-R values). Means, standard deviations, minimum, maximum, and correlations are presented in Table 1. There was a positive relationship between total psychopathy and primary psychopathy, *r*(95) = 0.883, *p* < 0.01, total psychopathy and secondary psychopathy, *r*(95) = 0.676, *p* < 0.01, and primary psychopathy and secondary psychopathy, *r*(95) = 0.251, *p* < 0.05. In addition, t-tests revealed that men had both higher levels of total psychopathy, *t*(89) = 2.288, *p* < 0.05, and primary psychopathy, *t*(89) = 2.712, *p* < 0.01, than women, though there was no gender difference between secondary psychopathy scores, *t*(89) = 0.453, *p =* ns.

As expected, total psychopathy scores were positively associated with utilitarian response rates (higher proportion of “yes” responses) in impersonal, *r*(93) = 0.239, *p* < 0.05, personal, *r*(93) = 0.197, *p* = 0.055, and overall moral dilemmas, *r*(93) = 0.269, *p* < 0.01. Primary psychopathy scores additionally correlated positively with utilitarian response rates in impersonal, *r*(93) = 0.223, *p* < 0.05, personal, *r*(93) = 0.312, *p* < 0.005, and moral dilemmas more broadly, *r*(93) = 0.352, *p* < 0.001. Secondary psychopathy scores were unrelated to the rate of utilitarian decision-making in any of the moral dilemmas. Reduced SCL-R during the contemplation period of personal moral dilemma trials correlated with increased utilitarian response rates for personal moral dilemmas, *r*(91) = −0.266, *p* < 0.05. There were no other relationships between utilitarian response rates, psychopathy scores, and physiological reactivity. Remaining correlations can be seen in Table 1.

Unexpectedly, the utilitarian response rates for impersonal moral dilemmas (M = 0.39, SD = 0.17), were significantly lower than for personal moral dilemmas (M = 0.51, SD = 0.28), t = −4.229, *p* < 0.001. However, a 2×1 ANCOVA showed that when controlling for primary psychopathy score, there was no significant difference between the two response rates, F(1,93) = 0.405, *p* = 0.526, ηp^2^ = 0.004. This was also true when controlling for total psychopathy score, F(1,93) = 0.066, *p* = 0.789, ηp^2^ = 0.001, though the difference remained significant when controlling for secondary psychopathy score, F(1,93) = 5.796, *p* < 0.05, ηp^2^ = 0.059. This finding suggests that primary and total psychopathy scores superficially inflated the “yes” response rates.

### 3.2. Mediation Analyses

We ran separate mediation analyses for each type of moral dilemma (impersonal, personal, and overall moral) as predicted by the three different psychopathic trait scores (total, primary, and secondary) for a total of nine models. While our hypotheses explicitly focused on primary psychopathic traits and personal moral dilemmas, we added the remaining mediation analyses post hoc for comparison. All models used the respective SCL-R score for each type of moral dilemma as a mediator. See Table 2 for mediation findings. Model values reflect point estimates of the posterior parameter distributions and 95% credible posterior density intervals. A credible interval that contains 0 suggests uncertainty in the distribution of the regression parameter and provides inconclusive information about the effect of the predictor. An analogous frequentist interpretation of a credible interval that includes 0 would be that the effect is not significant. We used *brms* default priors for model estimation, which are non-informative. Bayes Factors (BF10) greater than 3 offer at least moderate support for the alternative hypothesis, while those less than 1/3 offer at least moderate support for the null hypothesis.

SCL-R during personal moral dilemmas did not mediate the relationship between primary (θ = 0.0008, 95% CI (−0.0013, 0.0036)), secondary (θ = 0.0006, 95% CI (−0.0048, 0.0031)), or total psychopathy scores (θ = 0.0004, 95% CI (−0.0013, 0.0023)) and utilitarian response rates. However, there was a direct effect of primary psychopathy scores on personal moral utilitarian response rate (θ = 0.0121, 95% CI (0.0040, 0.0201)), indicating that high levels of primary psychopathic traits predicted increased utilitarian response rates. Low SCL-R during personal moral dilemmas was significantly related to increased utilitarian response rates (path *b*; all 95% CIs did not include 0). In addition, BF10 values were less than 0.33 when both primary and total psychopathy score were predictors, which suggests evidence in favor of the null hypothesis (that SCL-R does not mediate the relationship between psychopathic traits and utilitarian decision-making).

There were no significant indirect effects in any remaining models (Table 2).

## 4. Discussion

Consistent with past research and our first hypothesis, higher levels of psychopathic traits were associated with increased utilitarian responding during the moral dilemma task. This relationship was strongest for primary psychopathic traits, specifically, while secondary psychopathy trait score alone was unrelated to utilitarian response rate. This factor-specific effect is corroborated by prior findings including those of Tassy, Deruelle, Mancini, Leistedt, and Wicker (2013) [33] and Patil (2015) [56], who found that LSRP primary psychopathy score predicted utilitarian response rates, but not secondary psychopathy. These results and ours closely mirror those of Koenigs et al. (2012) [30], who used the Psychopathy Checklist Revised (PCL-R) [57], an interview-based clinical measure of psychopathy) in conjunction with the Welsh Anxiety Scale [58], and found that both high anxious (e.g., secondary) psychopaths and non-psychopaths were less likely to endorse utilitarian responses in personal moral dilemmas compared to low anxious (e.g., primary) psychopaths. The LSRP has demonstrated reliable overlap with the PCL-R—in particular, its capacity to evaluate both the affective and antisocial behavioral aspects of psychopathy [37,59]. In addition, the secondary factor of the LSRP correlates strongly with anxiety, as is typical of the antisocial behavioral facet of psychopathy [37]. Primary psychopaths have characteristically low levels of anxiety [5,60]; though the LSRP primary factor does not exhibit a strong negative relationship with anxiety [38], it indexes traits such as disinhibition and Machiavellian egocentricity that are typical of primary psychopathy [61].

Still, when comparing low-anxious psychopaths and non-psychopaths, Koenigs and colleagues (2012) [30] found a stronger relationship between levels of primary psychopathic traits and personal moral decision-making (Cohen’s *d* = 0.77) than we did. A median split of our sample along primary psychopathy score confirmed that although there was a significant difference between percent of “yes” answers elicited during personal moral dilemmas, the effect was not as robust (Cohen’s *d* = 0.44). However, Koenigs et al. (2012) conducted their study with a sample of incarcerated individuals, meaning their base rates of psychopathy were likely higher than those of our college sample, therefore differences dependent on psychopathic trait levels were likely intensified. Djeriouat and Trémolière (2014) [31] also found a stronger effect of psychopathy score on utilitarian responding (*r* = 0.38, a medium effect size) compared to our study (*r*= 0.27, a small-medium effect size). However, rather than having to make a yes/no decision, participants in that study specified the extent to which they would endorse a utilitarian action on a five-point scale compared to our forced-choice task. This may have resulted in participants that would otherwise have answered “no” in a forced-choice scenario offering some consideration of the utilitarian option. Patil (2015) [56] similarly used a Likert scale to ask participants whether or not the utilitarian action in a moral dilemma was appropriate; while they also found a relationship between primary psychopathy and utilitarian option endorsement during personal moral dilemmas, the strength of that relationship (odds ratio = 1.07) was roughly equivalent to a small effect size [62]. Likewise, Gao and Tang (2013) [2] found smaller effect sizes compared to our study for the association between total and primary psychopathy scores and utilitarian response rates to personal moral dilemmas, though they used the Psychopathic Personality Index (PPI) [63], a different measure of psychopathic traits in community samples.

Our skin conductance findings partially supported our second hypothesis, that SCL-R preceding a response during personal moral dilemmas would negatively relate to utilitarian response rates. This finding provides modest support for the harm aversion model, which posits that distress cues can aversively reinforce behavior due to increased physiological discomfort [56]. It also agrees with the findings of previous studies showing an association between blunted autonomic arousal and an increase in utilitarian decision-making during moral judgment [27,64]. However, SCL-R was notably unrelated to primary psychopathy in our study as predicted in the hypotheses. 

Many researchers attribute the utilitarian decision making associated with high levels of psychopathic traits in moral dilemmas to characteristic affective deficits and reduced emotional “interference” inherent to the primary psychopathy factor, in particular [61,65]. However, some recent research suggests that other factors may differentially influence moral decision making as a function of psychopathy. For example, Glenn and colleagues (2010) [7] found that individuals with high psychopathic trait levels were less likely to define their sense of self by their morality, which may partially explain their immoral behavior. In another study, a reduced concern for prosocial behaviors and an absence of moral values concerned with empathy and maintaining the physical integrity of others mediated the relationship between psychopathy and utilitarian decision making in sacrificial personal moral dilemmas [31].

Despite recent studies that provide evidence of a relationship between primary psychopathy and blunted skin conductance activity [66,67,68], as well as a comprehensive consensus that diminished neurological responding (including amygdala and hypothalamic-pituitary-adrenal axis activity) to negative/affective stimuli exemplifies primary psychopathy (for a review, see Yildirim and Derksen, 2015) [61], this study found no evidence of dampened physiological activity related to psychopathic trait levels. Nor was there a relationship between total psychopathy score more broadly and SCL activity. This contradicts existing evidence; men with high trait levels of PPI fearless dominance (roughly equivalent to primary psychopathy) showed a dampened skin conductance response to aversive images compared to those with lower fearless dominance scores [69]. Additionally, ventromedial prefrontal cortex (vmPFC) lesion patients (whose decision making mirrors that of psychopaths) during a moral dilemma task did not physiologically differentiate choosing the utilitarian option in a personal moral dilemma from not choosing the utilitarian outcome, while healthy controls and non-vmPFC lesion patients did [28]. Failure to find such relationships in our study may partly reflect the use of an undergraduate college sample, and not a clinical or incarcerated population. Further inconsistencies between research findings may also be due to comparisons across studies with different populations.

Recent research showed that, contrary to earlier theories, deficient affective processing was not the underlying mechanism for the observed increased utilitarian decision making in psychopathic individuals [29,70,71]. Conway et al. (2018) suggest that increased utilitarian decisions may be reflective of increased rational reasoning, rather than decreased emotional engagement [70]. Indeed, one of the limitations of sacrificial dilemmas is the inherent inability to distinguish the motivating factor behind a utilitarian decision, namely whether one is concerned with the greater good or has emotion processing deficits [29]. Similarly, the dilemmas used in the present study did not allow us to differentiate whether it was explicitly increased utilitarian or decreased deontological motivations that guided our participants’ actions. This is a limitation insofar as process dissociation is concerned. Lastly, as utilitarian endorsement in sacrificial dilemmas is associated with taking an action (e.g., flipping a switch or pushing a man), whereas the deontological decision is associated with inaction, a propensity toward action or inaction cannot be ruled out as a potential confounder [71]. De-confounding the propensity for action and underlying moral motivation in future studies is necessary and may increase the ability to detect underlying physiological mechanisms in moral decision-making [72,73].

Though psychopathic traits (primary score, in particular) and SCL-R were associated with utilitarian response rates, SCL-R did not mediate the relationship between psychopathy and utilitarian responding. Thus, while blunted sympathetic reactivity has been reported to be a core feature of (primary) psychopathy, our findings suggest that somatic marker production deficiencies may not be the underlying mechanism of characteristic utilitarian decision making in psychopathy. In the context of our study, it is possible that other affective traits such as risk taking or empathic concern contributed to the association between psychopathy and utilitarian moral judgment [61,74]; however, additional replication efforts should be made. Alternatively, the neural underpinnings that mediate the relationship between utilitarian decision-making and psychopathic traits in this sample may be seen in measures of structural and functional brain circuitry, rather than somatic marker production. Another possibility is that deficits in social-emotional behaviors (rather than physiological markers) reduced harm aversion and contributed to the greater endorsement of utilitarian decision making in personal moral dilemmas [75].

There are other possible explanations for the finding that participants with high levels of psychopathy did not show decreased somatic marker production preceding decision-making in personal moral dilemmas. For example, there may be a qualitative difference in how these individuals encoded physiological reactivity during the task; they could have experienced arousal in the form of pleasant affect at the idea of harming another [76], which resulted in selecting the utilitarian option. An alternative theory is that these individuals did experience physiological discomfort during the contemplation of personal moral dilemmas but failed to attend to and consciously process that feeling. For instance, the attention bottle neck theory proposed by Newman and Baskin-Sommers [77] argues that affective processing deficits in psychopathy largely arise when the emotional information in a scenario is peripheral to the main goal of the psychopath; thus, it is essentially ignored. By not attending to internal affective processing, impersonal and personal moral dilemmas are treated no differently.

This study offers an important addition to the literature on psychopathy and somatic marker production in nonclinical and non-forensic samples, providing findings that counter the harm-aversion and hypo-arousal theories of psychopathy in relation to utilitarian responding. Specific limitations should be noted, however, in the interpretation of this study’s conclusion. Namely, our sample consisted of undergraduate students (predominantly female) rather than clinical or incarcerated individuals. Further studies are needed to corroborate our findings in populations with higher base-rates of psychopathy; such complementary evidence would dispel a critical theory about reinforcement learning in the psychopathy/moral decision-making literature, and open the door to more innovative research exploring decision-making and arousal in this disorder.

## Figures and Tables

**Figure 1 brainsci-10-00303-f001:**
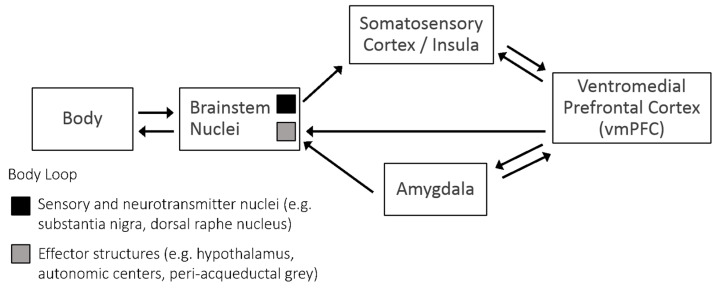
Sensory inputs enter the brain through the brainstem and are processed in the insula and somatosensory cortex. This information feeds forward into the ventromedial prefrontal cortex followed by the amygdala, which is involved in fear processing. Information feeds forward from the amygdala through the brainstem to autonomic centers resulting in increased arousal (like heart rate or skin conductance). The ventromedial prefrontal cortex (vmPFC) carries projections back to the somatosensory cortex and activates representations of sensory events, creating a hypothetical “as if” loop, in which a certain decision or event that has previously caused an unpleasant physiological response will activate effector structures via the vmPFC-amygdala pathway and recreate the unpleasant physiological response in the individual. Thus, as learning increases, anticipatory bodily responses start to precede decisions that might yield unpleasant outcomes. Modified from Bechara and Damasio (2005) [14].

**Figure 2 brainsci-10-00303-f002:**
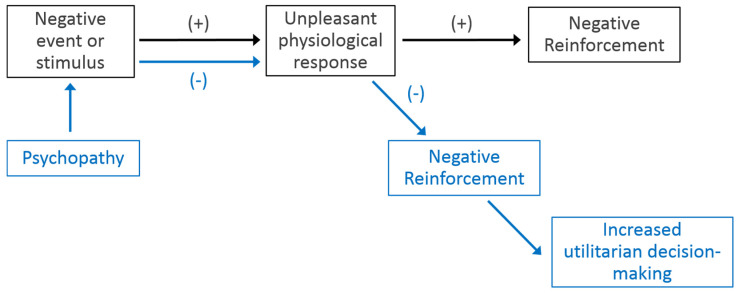
Contingency learning occurs when repeated exposure to a negative stimulus (top pathway) consistently results in an unpleasant physiological response (i.e., doing X makes me feel bad or causes pain). The increase in physiological response is proportionately related to increased learning. For those with high levels of psychopathic traits (bottom path), repeated exposure to a negative event (e.g., personally harming another person) fails to trigger a strong, unpleasant physiological response, purportedly due to insufficient autonomic arousal. Therefore, negative reinforcement of the negative event is not engaged, resulting in undeterred approach behavior.

**Table 1 brainsci-10-00303-t001:** Descriptive Statistics and Correlations Among Main Variables.

		1	2	3	4	5	6	7	8	9	10	11	12
1	Sex	1											
2	LSRP Total	−0.236 *	1										
3	LSRP Primary	−0.276 **	0.883 **	1									
4	LSRP Secondary	−0.048	0.676 **	0.251 *	1								
5	SCL-R Neutral	−0.14	−0.05	−0.065	0.008	1							
6	SCL-R Impersonal	0.145	−0.05	−0.129	0.096	0.291 **	1						
7	SCL-R Personal	−0.08	−0.05	−0.086	0.024	0.351 **	0.266 **	1					
8	SCL-R Moral	0.039	−0.07	−0.136	0.075	0.404 **	0.788 **	0.804 **	1				
9	Yes Neutral	−0.065	0.054	−0.022	0.146	0.077	0.034	−0.025	0.005	1			
10	Yes Impersonal	−0.021	0.239 *	0.223 *	0.142	0.186	0.039	0.074	0.071	0.016	1		
11	Yes Personal	0.077	0.197 †	0.312 **	−0.085	−0.15	−0.09	−0.266 **	−0.225 *	−0.229 *	0.204 *	1	
12	Yes Moral	0.05	0.269 **	0.352 **	0	−0.03	−0.05	−0.171	−0.14	−0.17	0.637 **	0.884 **	1
	Mean	1.71	50.2	29.51	20.69	−0.35	−0.19	−0.29	−0.24	0.8	0.39	0.51	0.45
	SD	0.46	9.16	6.97	4.44	0.47	0.32	0.33	0.26	0.15	0.17	0.28	0.18
	Minimum	1	29	18	10	−2.06	−1.33	−1.35	−1.11	0.4	0	0	0.1
	Maximum	2	71	47	32	0.57	0.41	0.37	0.26	1	0.8	1	0.8

† Correlation is significant at the 0.10 level (2-tailed); * Correlation is significant at the 0.05 level (2-tailed); ** Correlation is significant at the 0.01 level (2-tailed); LSRP = Levenson Self-Report Psychopathy scale; SCL-R = skin conductance reactivity during contemplation period of moral dilemma task; correlations in boldface type between behavioral, demographic, and physiological measures represent significant relationships between experimentally related variables (e.g., the correlation between skin conductance reactivity during personal moral dilemmas and the rate of harm endorsement during personal moral dilemmas).

**Table 2 brainsci-10-00303-t002:** The effect of skin conductance level reactivity on the relationship between psychopathic traits and utilitarian response rates.

Hypothesized Mediator (M)	Independent Variable (IV) Effect on *M (a)*		Association of M with Outcome Variable (b)		Direct Effect of IV on M (c’)		Indirect Effect		Proportion Mediated	BF10
Independent variable	Estimate	95% CI	Estimate	95% CI	Estimate	95% CI	Estimate	95% CI	%	
***SCL-Reactivity during impersonal moral dilemmas***										
Primary psychopathy	−0.0061	(−0.0158, 0.0039)	0.0355	(−0.0786, 0.1492)	0.0061	(0.0007, 0.0113)*	−0.0002	(−0.0014, 0.0006)	−2.04%	0.079
Secondary psychopathy	0.0076	(−0.0081, 0.0234)	0.0119	(−0.0977, 0.1245)	0.0056	(−0.0026, 0.0139)	0.0001	(−0.0012, 0.0014)	0.33%	0.149
Total psychopathy	−0.0016	(−0.0091, 0.0061)	0.0246	(−0.0872, 0.1362)	0.0047	(0.0007, 0.0086)*	0.0000	(−0.0007, 0.0005)	−0.22%	0.082
***SCL-Reactivity during personal moral dilemmas***										
**Primary psychopathy**	**−0.0041**	**(−0.0144, 0.0065)**	**−0.1913**	**(−0.3530, −0.0215)***	**0.0121**	**(0.0040, 0.0201)***	**0.0008**	**(−0.0013, 0.0036)**	**4.94%**	**0.217**
**Secondary psychopathy**	**0.0027**	**(−0.0132, 0.0186)**	**−0.2107**	**(−0.3774, −0.0403)***	**−0.0046**	**(−0.0177, 0.0087)**	**−0.0006**	**(−0.0048, 0.0031)**	**8.37%**	**0.375**
**Total psychopathy**	**−0.0018**	**(−0.0096, 0.0061)**	**−0.2037**	**(−0.3716, −0.0304)***	**0.0057**	**(−0.0008, 0.0121)**	**0.0004**	**(−0.0013, 0.0023)**	**4.47%**	**0.278**
***SCL-Reactivity during all moral dilemmas***										
Primary psychopathy	−0.0051	(−0.0132, 0.0031)	−0.0576	(−0.1979, 0.0832)	0.0091	(0.0036, 0.0143)*	0.0003	(−0.0006, 0.0016)	1.92%	0.117
Secondary psychopathy	0.0050	(−0.0070, 0.0171)	−0.0905	(−0.2443, 0.0568)	0.0007	(−0.0077, 0.0094)	−0.0005	(−0.0025, 0.0009)	−94.07%	0.917
Total psychopathy	−0.0016	(−0.0078, 0.0046)	−0.0795	(−0.2163, 0.0617)	0.0052	(0.0013, 0.0092)*	0.0001	(−0.0005, 0.0010)	1.16%	0.117

Note: CI = Credible intervals; BF10 = Bayes Factor; asterisks denote credible intervals where there is a 95% probability that the true value of the regression parameter estimate does *not* include 0, indicating that the predictor statistically influenced the outcome measure. Negative percentages in the *Proportion Mediated* column indicate that the indirect and direct effects had opposite directions and are not meaningful values. Bolded rows show Bayesian estimates from hypothesized models.

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
