# Peer review of "Somatic Marker Production Deficits Do Not Explain the Relationship between Psychopathic Traits and Utilitarian Moral Decision Making"

_brainsci, 2020, doi:10.3390/brainsci10050303_

Round 1

Reviewer 1 Report

I think this is a scientifically rigorous paper. It adds to the existing knowledge because it provides evidence contesting the thesis according to which somatic marker production underlies lower rates of utilitarian responding in persons with high psychopathic traits. The introduction describes accurately the goals of the research, the main literature on the topic is reviewed, the methodology used is adequate for the purpose described, a sufficient presentation of the methodology and the study conducted is provided, and both the conclusions and the limitations of the research are appropriately and coherently presented. Therefore, this is a paper I would suggest for its publication.

My only concern and critical remark is that it is based in my opinion on a solely biological explanation and comprehension of psichopathy. I would therefore suggest to complete it at least with some reference to studies which stress the influence of social and environmental factors on mental health and specifically on psichopathy. 

Author Response

The authors thank the reviewer for their feedback.

We added a sentence to the introduction indicating that social and environmental factors have also been implicated in the etiology of psychopathy (line 31). However, we are unsure if this is sufficient to address your comment. If you could please clarify which part of the Introduction we should expand on to incorporate the discussion of social and environmental factors, that will be immensely helpful.

Please note that per the suggestion of another reviewer, we added to the discussion section references that address factors that explain the psychopathy-utilitarian relationship that are not based on biological mechanisms.

Reviewer 2 Report

The study presented by the authors is a first test of the extent to which somatic marker deficits, i.e., skin conductance levels (SCL) explain the relationship between psychopathy and utilitarian decision-making in a college student sample. They hypothesized that higher levels of psychopathy would be related to increased utilitarian moral dilemma responses, that lower somatic responses (lower SCL) would be related to higher levels of psychopathy, and that the relationship between psychopathy and utilitarian response rates would be mediated by SCL. Replicating past research, they found that psychopathy was positively related to utilitarian responding, but psychopathy was not consistently correlated with somatic responses (correlation vs. mediation analyses).  For personal moral dilemmas only were lower SCLs correlated with utilitarian responses, and SCLs were not found to mediate the relationship between psychopathy and utilitarian responding.

Before proceeding with my review, I would like to note that my primary expertise is in the field of social psychology and thus my ability to evaluate the brain sciences, i.e., psychophysiology, portions of this submission are very limited. I may thus overlook important considerations related to the methods and findings in these fields, and thus rely on the other reviewers of this submission to thoroughly evaluate those aspects.

Overall, I very much enjoyed reading this paper, the writing was clear and concise, the study design and analyses were easy to follow. I think that the research question chosen by the authors is an important and interesting one, that has the potential to increase the depth of our understanding of moral psychology, i.e., the factors that influence our moral decision-making and their physiological underpinnings. I also appreciated the thorough acknowledgment of limitations in the discussion section.

I do think that the paper would benefit from addressing several issues before I would feel comfortable recommending it for publication:

  1. Some of the conclusions drawn by the authors seem to be overstated/not warranted by the results of their research. For example, in the abstract, “We conclude somatic marker production does not explain the connection between utilitarianism and psychopathy in a college population” and the title “Somatic marker production deficits do not explain the relationship between psychopathic traits and utilitarian moral decision-making”. This conclusion and the title seem pre-emptive given that the evidence consists of a single study, with a single measure of somatic marker production. Furthermore, “non-significant” results, are not strong evidence for the null hypothesis (e.g., Leppink, O’sullivan, & Winston, 2017). In addition, in regards to the mediation analyses, given that the results of such statistical modesl are still correlational and do not tell us about directionality, the authors should be careful about using causal language when describing the results (Sobel, 2008), although it is appropriate to discuss the causal direction that best fits with current theories.

  1. A) Important information to help the reader understand whether the study had sufficient power to detect effects was missing. How was the 100 participant sample size determined/calculated? What size of effects was the study powered to be able to detect?

B) Relatedly, many of the significant correlations presented were small in size, with p-values near the significance threshold (very few p < .001). How should readers interpret the size of these effects and their practical significance? For example, how do the effects in this study compare to those in other studies examining associations between personality traits and SCL, as well as between SCL and decision-making? How much more likely was a person in this study or (inferentially) is a person with an elevated level of psychopathy to make a utilitarian choice than someone with an average level of psychopathy?

Depending on word limits, such information could be presented in Supplemental Materials.

  1. In Table 1, the mean utilitarian response proportion is listed as .39 for impersonal and .51 for personal moral dilemmas, indicating that individuals chose the utilitarian option more often for personal than impersonal dilemmas? Is this maybe a typo or mistake? If not, it would be important to provide a discussion/explanation for this very surprising result.

  1. I appreciate that the authors acknowledge that the dilemmas that they used do not “allow us to conclude whether it is explicitly increased utilitarian rather than deontological motivations that guide our participants’ actions. This is a limitation insofar as process dissociation is concerned.” They then go on to state, “In the present study, however, this is more of a semantic issue that does not affect the question of whether somatic marker deficits underlie moral decision-making in those with high levels of psychopathic traits.” I would like to provide some pushback and suggest that the moral dilemmas used in this research may present more than a “semantic issue”. Because the set of dilemmas used in this study confound deontological responses with a general preference for inaction (and utilitarian responses with a general preference of action, since the utilitarian choice always requires taking action) a non-negligible proportion of variability in responses in the outcome variable is likely to be driven by (in)action tendencies and not feelings of harm aversion (see Gawronski, Armstrong, Conway, Friesdorf, & Hütter, 2017; Patil, 2015), thus potentially decreasing the sensitivity of the measure to detect moral decisions driven by harm aversion/a lack thereof, and thus the sensitivity of the mediation analysis (by decreasing the strength of the association between the IV and outcome, and M and outcome). If the relationship between the IV and mediator had been stronger in this study, then the use of more balanced dilemmas could have increased the likelihood of detecting a mediated effect. I would suggest that the authors discuss this potential limitation and how it might be addressed in future research.

Minor

  1. There are minor grammatical and language-related issues. For example, to the best of my knowledge “high psychopathic traits” are typically referred to has “high levels of psychopathy”, “high levels of trait psychopathy”, or “those high in psychopathy”. 

There are a number of spots with missing words (the) or issues with the plural form, e.g., “As expected, the total psychopathy score(S) had a positive relationship with the utilitarian response rate(s) (higher proportion of “yes” responses) in impersonal, r(95) = .239, p < .05, personal, r(95) = .197, p = 210 .055, and moral dilemmas overall, r(95) = .269, p < .01. Primary psychopathy additionally correlated 211 positively with the total psychopathy score had a positive relationship with the utilitarian response rate 209 (higher proportion of “yes” responses) in impersonal, r(95)utilitarian response rate in impersonal, r(95) = .223, p < .05, personal, r(95) = .312, p < 212 .005, and moral dilemmas more broadly, r(95) = .352, p < .001. Secondary psychopathy scores were unrelated to the rate of utilitarian decision-making in any of the moral dilemmas. Reduced SCL-R during the contemplation period of personal moral dilemma trials correlated with increased 215 utilitarian response rates for personal moral dilemmas, r(90) = -.249, p < .05. There were no other 216 relationships between utilitarian response rates, psychopathy scores, and physiological reactivity. The remaining correlations can be seen in Table 1.”

Author Response

Major

1) Some of the conclusions drawn by the authors seem to be overstated/not warranted by the results of their research. For example, in the abstract, “We conclude somatic marker production does not explain the connection between utilitarianism and psychopathy in a college population” and the title “Somatic marker production deficits do not explain the relationship between psychopathic traits and utilitarian moral decision-making”. This conclusion and the title seem pre-emptive given that the evidence consists of a single study, with a single measure of somatic marker production. Furthermore, “non-significant” results, are not strong evidence for the null hypothesis (e.g., Leppink, O’Sullivan, & Winston, 2017). In addition, in regards to the mediation analyses, given that the results of such statistical models are still correlational and do not tell us about directionality, the authors should be careful about using causal language when describing the results (Sobel, 2008), although it is appropriate to discuss the causal direction that best fits with current theories.

The authors agree with this comment and have adjusted the language in the abstract and in the discussion section. We also added a metric to the mediation analysis (Bayes Factors) that speaks to the concept of evidence in favor of the null and added those values to the mediation results table, as well.

2A) Important information to help the reader understand whether the study had sufficient power to detect effects was missing. How was the 100-participant sample size determined/calculated? What size of effects was the study powered to be able to detect?

We added information about a post-hoc power analysis to the Method section (line 203)

2B) Relatedly, many of the significant correlations presented were small, with p-values near the significance threshold (very few p < .001). How should readers interpret the size of these effects and their practical significance? For example, how do the effects in this study compare to those in other studies examining associations between personality traits and SCL, as well as between SCL and decision-making? How much more likely was a person in this study or (inferentially) is a person with an elevated level of psychopathy to make a utilitarian choice than someone with an average level of psychopathy?

The reviewer’s point is well taken, and we have therefore included more specific information in the discussion section about strength of relationship between psychopathic traits and decision-making in our study as compared to others (Line 309). Beginning line 331, we more fully contextualize our SCL-R findings within the current literature as well.

Depending on word limits, such information could be presented in Supplemental Materials.

3) In Table 1, the mean utilitarian response proportion is listed as .39 for impersonal and .51 for personal moral dilemmas, indicating that individuals chose the utilitarian option more often for personal than impersonal dilemmas? Is this maybe a typo or mistake? If not, it would be important to provide a discussion/explanation for this very surprising result.

The reviewer is correct in pointing out this oddity. We performed a paired-samples t-test to confirm that the difference between the personal and impersonal utilitarian response rates was statistically meaningful. After further probing, we discovered the difference between the two values was driven largely by psychopathy score. Specifically, after controlling for total psychopathy and primary psychopathy (separately), the difference between the two scores was no longer statistically significant. Both primary and total psychopathic trait levels inflated the ‘yes’ responses to personal moral dilemmas. We present the results of this brief analysis starting line 257. We felt it sufficient to not elaborate on this finding in the discussion but are happy to do so if the reviewer still thinks more explanation is warranted.

4. I appreciate that the authors acknowledge that the dilemmas that they used do not “allow us to conclude whether it is explicitly increased utilitarian rather than deontological motivations that guide our participants’ actions. This is a limitation insofar as process dissociation is concerned.” They then go on to state, “In the present study, however, this is more of a semantic issue that does not affect the question of whether somatic marker deficits underlie moral decision-making in those with high levels of psychopathic traits.” I would like to provide some pushback and suggest that the moral dilemmas used in this research may present more than a “semantic issue”. Because the set of dilemmas used in this study confound deontological responses with a general preference for inaction (and utilitarian responses with a general preference of action, since the utilitarian choice always requires taking action) a non-negligible proportion of variability in responses in the outcome variable is likely to be driven by (in)action tendencies and not feelings of harm aversion (see Gawronski, Armstrong, Conway, Friesdorf, & Hütter, 2017; Patil, 2015), thus potentially decreasing the sensitivity of the measure to detect moral decisions driven by harm aversion/a lack thereof, and thus the sensitivity of the mediation analysis (by decreasing the strength of the association between the IV and outcome, and M and outcome). If the relationship between the IV and mediator had been stronger in this study, then the use of more balanced dilemmas could have increased the likelihood of detecting a mediated effect. I would suggest that the authors discuss this potential limitation and how it might be addressed in future research.

We thank the reviewer for this thoughtful comment and we agree that this is an important limitation that should be considered when interpreting the results. We added this to our discussion (beginning line 363).

Minor

1) There are minor grammatical and language-related issues. For example, to the best of my knowledge “high psychopathic traits” are typically referred to has “high levels of psychopathy”, “high levels of trait psychopathy”, or “those high in psychopathy”. 

There are a number of spots with missing words (the) or issues with the plural form, e.g., “As expected, the total psychopathy score(S) had a positive relationship with the utilitarian response rate(s) (higher proportion of “yes” responses) in impersonal, r(95) = .239, p < .05, personal, r(95) = .197, p = 210 .055, and moral dilemmas overall, r(95) = .269, p < .01. Primary psychopathy additionally correlated 211 positively with the total psychopathy score had a positive relationship with the utilitarian response rate 209 (higher proportion of “yes” responses) in impersonal, r(95)utilitarian response rate in impersonal, r(95) = .223, p < .05, personal, r(95) = .312, p < 212 .005, and moral dilemmas more broadly, r(95) = .352, p < .001. Secondary psychopathy scores were unrelated to the rate of utilitarian decision-making in any of the moral dilemmas. Reduced SCL-R during the contemplation period of personal moral dilemma trials correlated with increased 215 utilitarian response rates for personal moral dilemmas, r(90) = -.249, p < .05. There were no other 216 relationships between utilitarian response rates, psychopathy scores, and physiological reactivity. The remaining correlations can be seen in Table 1.”

The authors thank the reviewer for pointing out these issues and have gone through the manuscript to make corrections.

Reviewer 3 Report

In the present manuscript, the authors report a study where they measured undergraduates’ level of psychopathy, their skin conductance, and their moral dilemma judgments. Psychopathy and utilitarian judgments as well as skin conductance and utilitarian judgments were correlated. However, the hypothesized mediation of psychopathy via skin conductance on utilitarian moral judgments did not occur.

Although I personally do not find the hypothesis very compelling, it is certainly a viable one and the study results are informative and should therefore be published. However, I think the treatment of the published literature on the association between psychopathy and utilitarian dilemma judgments needs to be substantiated. In the introduction, the discussion of previous morality research is largely confined to non-dilemma tasks; without clarifying how the mentioned literature transfers to dilemmas, where both options are wrong (which is why it is a dilemma), this does not seem very helpful to me. Although Greene’s dual process theory is sketched, it is unclear how psychopathy relates to this theory. Thus, my main criticism is that the present manuscript fails to discuss, let alone integrate, important research on the relationship between psychopathy and utilitarian moral judgments.

Prominent early research is missing, e.g., Bartels & Pizarro (2011; Cognition) and Kahane, Everett, Earp et al (2015; Cognition). Other research is also cited in a meta-analysis, by Marshall, Watts, & Lilienfeld (2018). More importantly, though, there is a substantial body of research that argues for alternative, that is non-emotional, mechanisms of how psychopathy might increase utilitarian judgments. Especially given the fact that the present authors do not find evidence for reduced emotional reactivity as a mechanism, a discussion of alternative mechanisms is, in my opinion, imperative. For example, Glenn, Koleva, Iyer, Graham, &Ditto (2010; JDM); Djeriouat & Trémolière (2014; Personality and Individual Differences); Körner, Deutsch, & Gawronski (2020; PSPB); Conway, Goldestein-Greenwood, Polacek, & Greene, 2018; Cognition); Balash, & Falkenbach (2018) and probably many more suggest that the link between psychopathy and utilitarian judgment need not be one of decreased affective processing.

Thus, although I think the study itself is soundly conducted (as far as I can judge), I think the introduction and discussion needs substantial improvements.

Minor point:

I did not like the mediation analysis table. First, I think it unnecessary to report 9 mediation analysis when there is really only a hypothesis for one of them (which could be nicely integrated into the main text). Second, in the current table, it took me quite a long time to understand which path was tested at which cell in the table. So, if Table 2 stays in the manuscript, I would recommend more explanations in the table note. Third, not being an expert on Bayesian mediation analysis, I do not think I understand what “proportion mediated” means if it can have a value of -94%. Thus, an explanation on that would also be advisable.

Author Response

In the introduction, the discussion of previous morality research is largely confined to non-dilemma tasks; without clarifying how the mentioned literature transfers to dilemmas, where both options are wrong (which is why it is a dilemma), this does not seem very helpful to me.

Although Greene’s dual process theory is sketched, it is unclear how psychopathy relates to this theory. Thus, my main criticism is that the present manuscript fails to discuss, let alone integrate, important research on the relationship between psychopathy and utilitarian moral judgments.

Prominent early research is missing, e.g., Bartels & Pizarro (2011; Cognition) and Kahane, Everett, Earp et al (2015; Cognition). Other research is also cited in a meta-analysis, by Marshall, Watts, & Lilienfeld (2018).

We thank the reviewer for this thoughtful feedback. We incorporated a discussion of prominent early research into our introduction in lines 108-110 and 111-123.

More importantly, though, there is a substantial body of research that argues for alternative, that is non-emotional, mechanisms of how psychopathy might increase utilitarian judgments. Especially given the fact that the present authors do not find evidence for reduced emotional reactivity as a mechanism, a discussion of alternative mechanisms is, in my opinion, imperative. For example, Glenn, Koleva, Iyer, Graham, &Ditto (2010; JDM); Djeriouat & Trémolière (2014; Personality and Individual Differences); Körner, Deutsch, & Gawronski (2020; PSPB); Conway, Goldestein-Greenwood, Polacek, & Greene, 2018; Cognition); Balash, & Falkenbach (2018) and probably many more suggest that the link between psychopathy and utilitarian judgment need not be one of decreased affective processing.

Thus, although I think the study itself is soundly conducted (as far as I can judge), I think the introduction and discussion needs substantial improvements.

We appreciate the reviewer’s feedback and agree that additional discussion of alternative mechanisms is warranted. We incorporated this in the Discussion section in lines 338-348 and 363-378.

Minor point:

I did not like the mediation analysis table. First, I think it unnecessary to report 9 mediation analysis when there is really only a hypothesis for one of them (which could be nicely integrated into the main text). Second, in the current table, it took me quite a long time to understand which path was tested at which cell in the table. So, if Table 2 stays in the manuscript, I would recommend more explanations in the table note. Third, not being an expert on Bayesian mediation analysis, I do not think I understand what “proportion mediated” means if it can have a value of -94%. Thus, an explanation on that would also be advisable.

The authors appreciate the reviewer’s comments about the mediation table and the suggestions about how to better clarify our results. We shaded the portion of the table that deals in the hypothesis-specific mediation results so as to highlight them for the reader, and added additional information about how to interpret the proportion mediated values in the table note. In addition, we added a column that includes Bayes Factors for each of the models. These values provide additional context to the interpretation of the results.

Round 2

Reviewer 3 Report

By including discussions of related research, especially in the Discussion, the authors removed my main concern.

However, reading the manuscript anew, I realized that the authors did not mention which specific dilemmas they used. They already cite the dilemma pool, so just additionally mentioning the names of the employed dilemmas should enable other researchers to replicate the procedure. I am sorry for not noticing this before.

Provided that the employed dilemmas are named, I advocate accepting the present manuscript.

Author Response

We thank the reviewer for this note. We added a comment to the Method section (line 183) to refer to Appendix A for the specific dilemma titles. The dilemma titles are subsequently listed in Appendix A.